# Public Health Implications and Possible Sources of Lead (Pb) as a Contaminant of Poorly Regulated Kratom Products in the United States

**DOI:** 10.3390/toxics10070398

**Published:** 2022-07-19

**Authors:** Walter Prozialeck, Alexandra Fowler, Joshua Edwards

**Affiliations:** 1Department of Pharmacology, College of Graduate Studies, Midwestern University, Downers Grove, IL 60515, USA; jedwar@midwestern.edu; 2Chicago College of Osteopathic Medicine, Midwestern University, Downers Grove, IL 60515, USA; alexandra.fowler@midwestern.edu

**Keywords:** lead, kratom, environmental toxicology, opioids, mytragynine

## Abstract

Kratom (*Mitragyna speciosa*) is a tropical tree that is indigenous to Southeast Asia. Kratom leaf products have been used in traditional folk medicine for their unique combination of stimulant and opioid-like effects. Kratom is being increasingly used in the West for its reputed benefits in the treatment of pain, depression, and opioid use disorder (OUD). Recent studies from the United States Food and Drug Administration (FDA, Silver Spring, MD, USA) and our laboratory have shown that many kratom products being sold in the United States are contaminated with potentially hazardous levels of lead (Pb). In this commentary, we discuss the public health implications of the presence of Pb in kratom products, particularly as they relate to the predicted levels of Pb exposure among kratom users. We also considered the specific toxic effects of Pb and how they might relate to the known physiologic and toxicologic effects of kratom. Finally, we consider the possible sources of Pb in kratom products and suggest several areas for research on this issue.

## 1. Introduction

Kratom (*Mitragyna speciosa*) is a tropical tree that is indigenous to Indonesia, Thailand, Malaysia, and other regions of Southeast Asia [1,2,3,4]. Native populations in these regions have used kratom leaves or decoctions prepared from the leaves as a mild stimulant to stave off fatigue and as an opioid substitute to manage pain or treat opioid use disorder (OUD) [5,6,7]. Pharmacologic studies have revealed that kratom leaves contain over 40 active alkaloids, with the best-known being mitragynine [8,9,10]. Mitragynine acts as a partial agonist at mu-type opioid receptors [8,9,10] and can modulate the activity of neurotransmitters such as norepinephrine and serotonin [8,11,12]. 

The general subjective effects of kratom have been summarized in a previous review [3]. Kratom produces an unusual combination of stimulant and opioid-like effects. These effects are highly dependent on the dose of kratom and can vary greatly from one person to another. Low to moderate doses (1–5 g raw leaves) usually produce a mild stimulatory effect that most people perceive as pleasant but not as intense as those of amphetamine-like drugs. Some individuals, however, have reported that low-dose effects of kratom are characterized by an unpleasant sense of anxiety and agitation. It is noteworthy that those who have used kratom products for pain management tend to view the stimulant effects as being more desirable than the sedative effects of traditional opioids.

Opioid-like effects such as analgesia, constipation, euphoria, and sedation are typically associated with the use of moderate-high doses of kratom (5–15 g). As with the low-dose effects, the higher-dose effects may be either euphoric or dysphoric, depending on the user. Of note, the euphoric effects of kratom tend to be less intense than those of opium and opioid drugs. Regardless, kratom continues to be sought by illicit drug users. Very high doses of kratom (>50 g) produce variable effects that are mainly excitatory (tachycardia and seizures).

While kratom has been used in Southeast Asia for hundreds of years, it is only within the past 20 years that kratom use has expanded to Europe and the Americas [1,4,5,13,14]. In the United States, kratom products are usually used for the self-management of pain, opioid use disorder, and depression [13,14,15,16,17,18,19]. The most widely used kratom products include finely ground dried leaf powder (either alone or formulated into capsules or tablets) or concentrated leaf extracts [3,4,16]. These types of products are available from Internet vendors or specialty stores commonly known as “head shops” or “smoke shops”.

As kratom use has increased in the United States, there have been growing concerns about the safety of kratom products. Much of the concern has focused on the increase in the number of reports to poison control centers in which kratom was mentioned as a contributing factor in the poisoning scenarios including several deaths [16,20,21,22,23]. Such reports of adverse effects were cited by the United States Drug Enforcement Administration (DEA, Springfield, VA, USA) in a proposal to place kratom and its mitragynine constituents into Schedule I of the Controlled Substance Act [24], an action that would have effectively banned the possession and sale of kratom products in the USA. However, definitive proof that kratom was the cause of the toxicities and deaths was lacking. In addition, almost all cases involved the use of other drugs or the presence of serious health problems including seizure disorders and refractory depression [16,25,26,27]. In some cases, the kratom products may have even been adulterated with exogenous substances including synthetic opioids [16,28,29]. The DEA’s proposal to ban kratom prompted a massive response by kratom advocates that led the DEA to place the scheduling of kratom and its mitragynine constituents on indefinite hold [23]. In October 2021, the World Health Organization’s (WHO, Geneva, Switzerland) Expert Committee on Drug Dependence (ECDD) met to consider a proposal to move kratom and its mitragynine to Schedule I status at the level of the WHO. The ECDD voted against proceeding with the proposal [30]. Thus, as of the time of this writing, kratom remains legal throughout most of the world, although it is banned in several nations including Australia, Denmark, Finland, Israel, Japan, Malaysia, New Zealand, Poland, Romania, Russia, Singapore, South Korea, Sweden, and Vietnam.

Over the past three years, additional safety concerns have been raised about kratom products. One of the major issues has focused on the presence of toxic metals such Ni and Pb in many kratom products being sold in the USA. In a report posted in April 2019, researchers at the Food and Drug Administration (FDA, Silver Spring MD, USA) analyzed 30 kratom products for the presence of Ni and Pb. They found that most products contained significant levels of Ni, and all but two of the products contained relatively high levels of Pb [31]. In a follow-up study, Prozialeck et al. [32] analyzed eight kratom products purchased from “head shops” in the Western suburbs of Chicago and found that six of the products contained relatively high levels of Pb, Ni, Fe, and Cr, but only trace levels of several other metals including As, Cd, and Hg. Trace levels were defined as those that were below the limit of detection of the methodology utilized (inductively coupled plasma mass spectrometry) or well below the levels that would be considered as health risks to consumers.

A critical question, of course, is, do any of these metals represent a hazard to kratom users? As noted in their report, Prozialeck and coworkers [32] concluded that of all the metals that were detected, the one that most likely posed the greatest hazard to humans was Pb. Their data, along with that of the FDA, showed that levels of Pb in many kratom products were so high that kratom consumers using even moderate doses of kratom could easily exceed the daily threshold levels of Pb intake.

In this commentary, we discuss the public health and pharmacologic implications of the presence of Pb in kratom products in the United States, consider the possible sources of Pb in kratom, and suggest several areas for future research regarding this matter.

## 2. Overview of Pb as an Environmental Health Problem

Pb is a widespread pollutant that is currently ranked #2 on the United States Environmental Protection Agency’s (EPA, Washington, DC, USA) Toxic Substances and Disease Registry List of Hazardous Substances [33]. In addition, the FDA has classified Pb as a Class I contaminant [34], which reflects the fact that Pb has the potential to cause serious toxicities, especially with chronic exposure. Pb can cause neurological, psychological, cognitive, behavioral, reproductive, developmental, immunologic, cardiovascular, and renal effects [33]. These effects most likely result from the ability of Pb to induce oxidative stress and mimic or antagonize the effects of essential metals such as Ca, Fe, and Zn [33]. The CNS effects may be especially serious or even fatal in children aged 5 and younger due to the damaging effects of Pb on the blood–brain barrier [35]. Pb also causes direct neuro-inflammation in pre and neonatal rat pups [36]. For young children with Pb exposure, these CNS effects lead to poor clinical outcomes such as lower IQ, attention deficit hyperactivity disorder, seizures, coma, and death. As such, it is now widely accepted that there is no absolute “safe” level of exposure of Pb for children aged 5 and younger. As new information on the damaging effects of Pb on the developing CNS has emerged, the threshold or level of concern blood lead levels (BLL) have subsequently decreased. Prior to 2012, the level of concern BLL for children was 10 µg/dL; from 2012 to 2021,it was 5 µg/dL; and since 2021, it has been 3.5 µg/dL [37]. In adults, like children, the adverse health effects of Pb exposure is an area needing further study. However, it is known that a BLL of 10 µg/dL and below may cause decreased hemoglobin synthesis resulting in hypochromic microcytic anemia; increased systolic and diastolic blood pressure; altered bone formation, immunotoxicity, GI distress, and pain as well as behavioral effects and damage to the reproductive system (for review see [38]). The autonomic system is also a target of Pb toxicity [39]. Renal damage resulting in proteinuria and polyuria occurs at elevated levels of 60 µg/dL BLL and the inhalation of Pb dust may cause decreased respiratory system function [38]. These effects occur at such a low BLL that the EPA does not have an official reference dose, the maximum daily oral exposure, which is anticipated to not result in an appreciable health risk for Pb [37].

In considering kratom as a potential source of Pb exposure, it is important to note that young children are far more sensitive than adults to the adverse effects of Pb. At the same time, it is clear that post-adolescents and adults are far more likely than children to consume kratom. Nevertheless, children might be accidentally exposed to kratom or may even be given kratom by their parents or guardians for the treatment of various ailments. Such use of any kratom product in children would be highly inappropriate. Moreover, the possibility that kratom or Pb in the kratom products might affect developing children is a very important factor in discouraging the use of kratom in pregnant or lactating women, and even women who are trying to become pregnant. All kratom products, but especially those containing Pb, should be avoided in such populations. 

While it is widely accepted that no level of Pb exposure is truly safe, especially for children, Pb is ubiquitous in our environment, water, and food supplies. The FDA has established action levels for concentrations of Pb in water and various foods, but the levels are variable and based on the expected level of intake of the various materials. In 1992, the FDA established the provisional tolerable total daily intake (PTTDI) level of Pb from foods at 75 µg/day for adults [40]. With regard to drugs/pharmaceuticals, the FDA has an established a permitted daily exposure (PDE) level for Pb of 5 µg/day [34] and also established a new interim level (IRL) of 12.5 µg/day for total Pb exposure in adults [41]. Even though kratom is technically classified as a dietary ingredient, in reality, it is almost always used in the same manner as a drug or pharmaceutical [42,43]. Hence, our discussion of the significance of Pb in kratom will focus on the PDE level of 5 µg/day and an IRL of 12.5 µg/day.

## 3. Legal Status of Kratom and Overview of the Kratom Industry in the USA

To consider the possible significance of the presence of Pb in kratom, it is first necessary to consider the current legal status of kratom and the environment in which the kratom industry operates. In the USA, kratom is regarded as a herbal supplement under the FDA and DEA and is not regulated tightly as are pharmaceuticals. Although it remains legal in most of the USA, at the time of writing, several states such as Alabama, Florida, Indiana, Arkansas, Wisconsin, and Tennessee have passed legislation banning the local sale and possession of kratom. However, several states have also passed or are considering so called kratom consumer protection acts, which allow for the sale and use of kratom, but also establish standards for the quality control of kratom products. In addition, the American Kratom Association (AKA, Haymarket, VA, USA) an advocacy group founded by kratom consumers, has developed a set of good manufacturing practices (GMP) that directs kratom producers to adhere to quality control standards that include the testing of products for levels of toxic metals including Pb that are in line with the FDA policies for food stuffs [44]. Kratom producers are encouraged to agree to adhere to the GMP standards but are not required to do so. As a result, there are many kratom vendors in the USA who sell poor quality products that may contain toxic metals [31,32]. In addition, some unscrupulous producers and vendors sell products that may be adulterated with other psychoactive drugs including synthetic opioids such as fentanyl [16,21], or include plant materials other than kratom [45]. In this poorly regulated environment, it is not surprising that many of the kratom products being sold in the USA might contain hazardous levels of Pb. It is also not surprising that many kratom producers and vendors have been less than forthright in providing information about the sourcing and processing of their products [46,47]. This problem has been further compounded by the fact that the U.S. government has recently issued import alerts and seized shipments of kratom from Indonesia [48].

## 4. Potential Levels of Pb Exposure in Kratom Users

While definitive numbers on the extent of kratom use in the USA are not available, estimates based on online surveys suggest that there are between 2–4 million kratom users in the USA. Other estimates based on kratom import data and typical doses place the number of users as much higher at 10–15 million (for a review see [42]). In the USA, the maximum daily exposure level for Pb in pharmaceuticals is only 5 µg/day [34]. The results of studies from the FDA [31] and Prozialeck et al. [32] showed that levels of Pb varied across samples of kratom. While a few samples contained little or no Pb, most samples in both studies contained levels of Pb that ranged from about 0.2 µg/g of raw leaf material to over 1.4 µg/g. These levels are significant and could represent a major source of Pb exposure for kratom consumers. Doses of kratom vary among individual users. While many users report taking low doses of 2–5 g/day, it is not uncommon for heavy users to consume 10–60 g/day, or even higher [42].

Figure 1 shows the range of potential levels of Pb exposure (bars) as a function of kratom dose (g/day) in samples that were analyzed by the FDA [31]. The crossed lines in the graph illustrate the PTTD at 75 µg/day, IRL at 12.5 µg/day, and the PDE for pharmaceuticals at 5 µg/day. As can be seen, the level of Pb intake from kratom can easily surpass the FDA-defined limits. This level of Pb exposure could represent a major hazard for kratom users. It is also important to note that the vast majority of kratom users would probably be exposed to Pb from other sources (food, drinking water, etc.) Thus, even if the intake of Pb from kratom does not exceed the 5 µg/day threshold for pharmaceuticals or the 12.5 µg/day IRL for total intake, the use of kratom could still contribute significantly to the total Pb exposure. The data in Figure 2 highlight this point. The figure shows the levels of Pb in several common foods compared to the mean level of Pb found in the 30 kratom products analyzed by the FDA [31]. While the total intake of Pb would depend on the amount of food or product consumed, the data indicate that kratom could be a major source of Pb exposure. For example, a modest serving of fruit cocktail would contain 100 g of product and a chocolate bar might contain about 40 g of product. If a person consumed both products, the total lead intake would be about 1.4 µg. If the person also consumed a modest 5 g dose of kratom, they would be ingesting another 2.5 µg of Pb. It is also important to note that heavy kratom users may take doses as high as 50–100 g of kratom, which could include very high amounts of Pb.

It is also important to note that Pb is just one of many toxic metals to which individuals might be exposed. These other metals include Ni, Cr, and Cd, which are all known to be present in kratom as well as in the diets of many Americans. Lifestyle factors such as smoking can represent a source of exposure, especially to Cd. While little is known about how Pb interacts with other metals, there is evidence that Pb can enhance the nephrotoxic effects of metals such as Cd [49]. A recent case report from [50] highlighted the development of Cd-induced Fanconi syndrome in a subject who had been using high doses of kratom. However, the authors did not determine the levels of Pb in the subject.

## 5. Possible Role of Pb in “Kratom” Toxicity

The presence of Pb in kratom could increase the risk of Pb poisoning, especially in chronic users of kratom products. However, the presence of Pb in kratom also raises several other questions. An obvious question is, might Pb be a contributing factor in some of the increasing numbers of reports of kratom poisoning incidents in the USA? In considering this issue, it is important to note that even though kratom teas and other decoctions have been used by peoples of Southeast Asia for generations, there have been few reports of kratom toxicity or kratom-related Pb or poisoning from that region of the world [16,42]. It is only over the past 15–20 years when kratom usage spread to Europe and the USA that reports of toxicities began to appear in the literature [16,42]. In addition, the adverse effects of kratom products in the West appear to be qualitatively different and more severe than the effects reported among traditional users in Southeast Asia [51].

Even though kratom and its mitragynine constituents are often referred to as opioids or atypical opioids, the toxic effects that have been reported for kratom overdose cases in the USA are quite different from the toxic effects that are commonly associated with opioid overdoses. For example, opioids tend to depress the CNS, respiratory, and cardiovascular function. In contrast, recent analyses of reports of kratom toxicity in the USA have indicate that the most common adverse effects include nausea, vomiting, CNS excitation, seizures, arrhythmias, hypertension, hepatic injury, and nephrotoxicity [52,53,54]. Interestingly, Pb is well-known to damage many of the same organs that seem to be affected by kratom such as the brain, liver, kidney, GI tract, and heart [33,43]. It is also important to note that while we have focused the present discussion on the potential adverse effects of chronic Pb exposure from long-term use, it is also possible that Pb may contribute to some of the effects that have been reported to occur with acute kratom overdose situations. Table 1 summarizes some of the most prominent effects of kratom and Pb on key organ systems. As can be seen, both kratom and Pb have been reported to target many of the same systems (central nervous system, peripheral nervous system, gastrointestinal system and liver). This raises the possibility that some of the toxic effects that have been attributed to kratom may have been caused in part by Pb in the kratom products. In considering this issue, it is important to note that it is not uncommon for heavy kratom users to consume doses of over 50 g, which could contain 300 µg of Pb. Such high doses of Pb could contribute to some effects such as GI distress, hepatic and renal injury, which have been attributed to the use of kratom products in the U.S.

In addition to the possibility that Pb might cause or contribute to some of the effects that have been reported with kratom overdoses in the USA, there is also the possibility that Pb might interact at the mechanistic level with the mitragynines and other active constituents of kratom. We found no reports in the literature that address this issue and the topic would certainly merit further attention. However, this would be an incredibly complicated issue to address because Pb is a non-selective toxicant that can induce oxidative stress, affecting almost every organ system. Kratom leaves are known to contain over 40 active alkaloids and little is known about their molecular actions, but the possibility that Pb might influence the actions of alkaloids found in kratom is a topic that merits further study.

## 6. Potential Sources of Pb in Kratom

An obvious question that arises is, what is the source of the Pb in kratom products? While we can only speculate at this time, one possibility is that Pb might bioaccumulate from the soil and water where kratom is grown. Kratom usually grows in very wet, tropical environments, often on the flood plains of small rivers. While we found no data on the bioaccumulation of metals by kratom, there is a large volume of information indicating that other plants such as mangrove and cacao, which also grow in water-rich environments, can bioaccumulate metals. It is noteworthy that over 90% of the kratom products sold in the USA originate from sources in Indonesia [4,55,56]. The volcanic soil in many regions of Indonesia is known to contain high levels of metals [57]. In addition, Pb pollution is a widespread problem in some regions of Indonesia [58,59].

In considering the possible association between the geographic source of kratom products and levels of Pb, a recent study by Braley and Houdrogiannise [46] could provide important insights. In that study, the authors attempted to correlate the levels of a panel of essential and nonessential metals with the purported country of origin of kratom products purchased from an online vendor in the USA. Their results showed that Pb was present in all samples, although the levels of Pb were significantly lower than those reported by the FDA [34] and Prozialeck et al. [32]. However, their data suggest that there may be different levels of Pb in kratom grown/sourced from various countries. The highest levels of Pb were found in samples that had purportedly been produced in “Indonesia,” which is the source of most kratom that is shipped to the USA. Interestingly, the samples that supposedly originated from “Borneo”, which is largely controlled by Indonesia, but also includes regions controlled by Malaysia and Brunei, showed significantly lower levels of Pb than those sourced from “Indonesia”. It should be emphasized that the authors of that study could not verify that the kratom products had actually been sourced from the countries of the purported origin. Nevertheless, this would certainly seem to be an area that merits further research.

Another potential source of Pb could be the water that is used in the washing and processing of kratom leaves. After the leaves are harvested, they are washed and then prepared for “drying”. In many instances, the harvesting, drying, and processing are conducted in remote areas of the world that might not have access to high quality potable water. After washing, the kratom leaves are subjected to a “drying” process but are not totally desiccated. As noted by Prozialeck et al. [32], even “dry” kratom leaf products still contain 10–15% water by weight. It is important to note that any drying process would not remove the Pb that may have been in the water used to wash the kratom. In fact, drying would concentrate the Pb in or on the kratom leaves. 

The third possible source of Pb could be the equipment that is used to grind, process, and store the kratom leaf material. While there are certainly some producers who use modern, state-of-the-art equipment that would not be a source of Pb, there are many other lower-level producers who use primitive food processing equipment in kratom production. Such equipment, whether for grinding, storing, or shipping, could be a significant source of Pb contamination. While information is lacking on Pb levels during the processing of kratom leaves, studies have been conducted to determine the source of elevated Pb content in cocoa (*Theobroma cacao*) products and the machinery used to process the cocoa products. One study from East Luwu, South Sulawesi, Indonesia found elevated levels of Pb in the cocoa shells at 5 to 11 mg/kg, but the cocoa beans had undetectable levels [60]. This study demonstrates the ability to concentrate Pb in certain areas within the body of the plant. In another study from Africa examining the Pb content of cocoa products, data indicated that Pb leached from Pb-containing industrial machinery or storage containers used in the manufacture of cocoa products [61].

## 7. Conclusions

The results of the studies described in the present report indicate that many kratom products being sold in the USA contain levels of Pb that could pose a significant health risk to kratom consumers. However, there are many aspects of this issue that require further research and attention. First, studies to identify the source or sources of Pb are urgently needed. An important first step could be to trace the levels of Pb during kratom growth and production (i.e., from the soil to the market). A second area of research could focus on the possible bioaccumulation of Pb by the kratom plant (*Mitragynine speciosa*). Can the plant take up Pb and other metals from the soil and water, and what are the factors that determine the level of uptake (e.g. pH, age of plant, microbiome, etc.)? It would also be extremely important to determine how levels of Pb in kratom might correlate with the geographic regions in which the plant is grown. Finally, it would be interesting to determine how kratom might be grown and processed in ways that minimize the risks of Pb contamination.

We have advocated for further research on the therapeutic potential of kratom for the treatment of pain and as a harm-reduction agent in the current opioid crisis [16], and we stand by that position. However, we also find the present findings to be troubling. It is apparent that many of the kratom products being sold in the USA are contaminated with potentially dangerous levels of Pb. This puts consumers at potential risk of adverse effects. Even though the AKA has adopted and advocated GMP and some states are passing kratom consumer protection acts, it is obvious that many purveyors of kratom products have not adopted or adhered to these standards. It is our hope that the present analysis will serve as a template for more extensive studies on the large number of poorly-regulated kratom products that are being sold in shops and through Internet vendors. Such data are critical in formulating rational standards for the sale and production of kratom products.

## Figures and Tables

**Figure 1 toxics-10-00398-f001:**
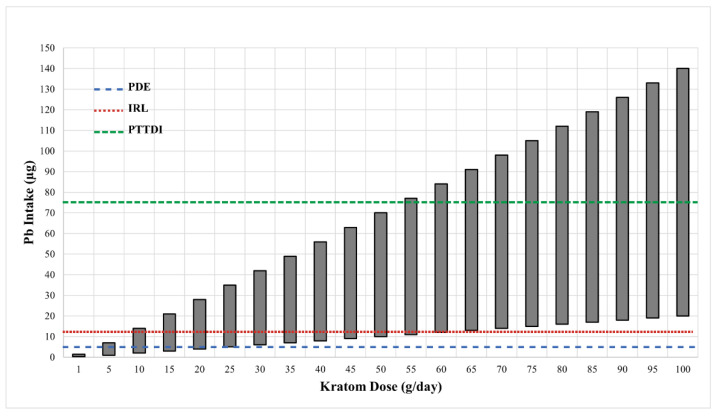
The estimated daily intake of Pb from various doses of kratom. Data on the ranges of Pb levels in 28 kratom products were obtained from the FDA [31]. The Pb values in those samples varied from 0.2–1.4 µg/g kratom. The vertical bars show the range of Pb intake at each of the kratom doses shown on the X-axis. The dotted horizontal lines show the various FDA target goals for Pb intake. The line at 75 µg represents the FDA’s provisional tolerant total daily intake (PTTDI) from food. The line at 12.5 µg represents the FDA’s interim reference level (IRL) from food, and the line at 5 µg represents the permitted daily exposure (PDE) from pharmaceuticals.

**Figure 2 toxics-10-00398-f002:**
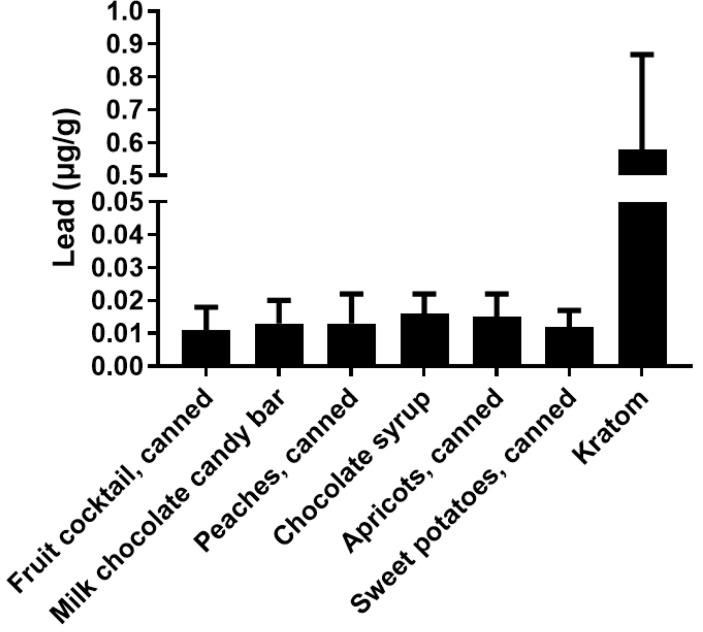
A comparison of the levels of Pb in various foods and kratom. The results show the mean ± SD for levels of Pb (µg/g) from various foods and kratom. The kratom data were obtained from the FDA (2019). Data for the other materials were obtained from the FDA Total Diet Study.

**Table 1 toxics-10-00398-t001:** A summary of the major pharmacologic/toxicologic effects of kratom and Pb.

Organ System	Kratom	Pb
Central Nervous System	Dose-dependent: stimulation, analgesia, sedation and seizures	Behavioral changes, developmental and cognitive impairment, ADHD, seizures, encephalopathy
Peripheral Nervous System	Tremors	Extensor muscle neuropathies and tremors
Gastrointestinal System	Constipation	Pain and colic
Hepatic	Multiple forms of injury	Multiple forms of injury
Cardiovascular	Arrythmias	Cardiomyopathy and hypertension
Renal	Unknown	Decreased glomerular filtration rate and tubular injury
Hematopoietic	Unknown	Microcytic hypochromic anemia
Skeletal	Unknown	Main site of accumulation; bone loss and periodontal disease

The effects are summarized from references [3,14,16,22,33].

## Data Availability

Data available upon request.

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
