# Peer review of "Public Health Implications and Possible Sources of Lead (Pb) as a Contaminant of Poorly Regulated Kratom Products in the United States"

_toxics, 2022, doi:10.3390/toxics10070398_

Round 1

Reviewer 1 Report

This is an interesting communication on the potential for Pb exposure from kratom use.  I'm not entirely convinced that the concern is warranted, but certainly further study is needed.  There were some areas I thought could be filled out better, and some of my specific questions and concerns are detailed below.  

There’s a lot of use of quotation marks that doesn’t really make sense.  I can see using it for colloquialisms like “head shop” but why is kratom sometimes in quotation marks and other times not?  Why is poisoning in quotation marks?  Why are safe and action levels in quotation marks? 

1.      Introduction:

P2, L64-69:  Do you have more specific information on what “significant levels,” “high levels,” and “trace levels” are?  What were these determinations based upon?  (Does trace mean detectable but not quantifiable?  Does high mean above some FDA limit or a given analyte-dependent value?)  Also, maybe the term “concentration” is more specific than the term “level.”  I am especially curious because the following paragraph states that most of the metals were not in high enough concentration to be of concern to consumers.

2.      Overview of Pb:  Much of the discussion is on childhood exposures, which is fair because that’s been the public health focus, but children probably won't be the primary kratom users.  You do touch upon effects on adults.  Then you discuss concentration limits for food and drug products.  I think it might be worth emphasizing that kratom exposure is more likely to occur in adolescent or adult people.  Do you have any special concern for women of reproductive age, or women who are pregnant or nursing?

P2, L83; P3, L101; P5, L187 & 194:  “toxicities”: the dictionary definition of toxicity is the “quality, state, or relative degree of being poisonous” even though the word is often used to describe a clinical condition.  Perhaps poisoning, intoxication, overdose, or some other words would be more suitable in some places? 

P2, L84:  Slight disagreement with your wording. I would say that chronic Pb exposure is certainly more common in most of the settings of concern here, but acute exposure is equally serious, if less insidious. 

3.      Legal & overview of kratom:

P3, L122: I’m not sure of the convention for this journal, but in general you don’t have to use parenthesis for FDA and DEA once you have defined the abbreviations.

P3, L129-131:  Can you give more specific information on product testing and allowable heavy metal concentrations based on the AKA GMP recommendations? 

P3, L133-135:  Is there a difference in labelling between products using AKA GMP recommendations versus products that do not?  Out of curiosity, is there any way of knowing that the products actually contain kratom?  It looks like one of your references addressed this question.

4.     Potential Levels of Pb:

P3, L146-147:  maximum daily exposure limit?

P3, L147-148:  I would specify “samples of kratom.”  I had to reread a couple times because the previous sentence references pharmaceuticals in general.

P4, L150:  Just requesting clarification: the raw leaf material is unprocessed and not powdered?  As in, whole leaves?  Is this value dry matter, fresh, or as packaged?  Does the 1.4 micrograms/g reference raw leaf material, packaged material, or all kratom samples?  Do both studies use either dry matter or fresh weight (and which do they use)?

P4, L156-158:  Sentence structure issues, and there is a closed parenthesis missing.

P4, L159:  Sentence ends “of.”

P4, L61-166:  You need to better explain why it's appropriate to compare food values to pharmaceutical values, since it isn't clear to me that it is a reasonable comparison.  So, a serving size container of fruit cocktail is little over 100 g, and a chocolate bar is about 40g.  That would mean a single serving of either would have around 1 microgram Pb.  If the average kratom user says they take 5g doses, that would be exposed to about 2.5 micrograms. 

P5, L184: omit “from” before reference 49 insertion.

5.      Possible role of Pb in kratom (overdose):  Based on the exposure levels from kratom and the known Pb concentrations, are the potential Pb doses from kratom consistent with the doses that would be needed to cause the organ damage being described?  The list of organs affected by both is, honestly, not unexpected as many toxicants affect multiple organ systems through various pathways, chemical forms, etc (a fact you summarize on P6). 

P5, L203-204: The previous sentence refers to overdose of opioids.  Are the adverse effects described here due to an acute overdose, or chronic use, or some combination thereof?  How do they compare to the overdose effects that were described in earlier studies of more traditional Thai users?

6.      Potential sources of Pb: has any work been done to see if kratom used in traditional settings is contaminated?  You’d mentioned the potential for fraud (spiking kratom with fentanyl, for example) early in the paper.  Is there any potential for substituting some other (plant or not plant) material for kratom that could be influencing the Pb content?

P7, L260-621:  I’m not sure why the information on location of Pb in cocoa pods is pertinent here, though it is interesting.  (It also could be due to surface contamination and not necessarily plant physiology). 

7.      Future studies

P7, L268: although I agree with the next sentence about further attention, and I am of the opinion that any exposure is too much exposure, I’m still not convinced that the Pb in kratom poses a “significant health risk” at the concentrations mentioned herein.  Especially since this is mostly an adult exposure problem, and Pb is far more bioavailable and toxic to children.  Though I think it might be worth considering child-bearing women using kratom as potentially problematic.

P7, L271:  I agree that determining the source of Pb in kratom products is critical!  I would also emphasize the processing as a potential contamination source, and the potential for fraud which could also conceivably contribute to the contamination issue. 

P7, L280-281:  Agree with you here, based on my limited knowledge of kratom, I think it could be a beneficial drug for palliative care and harm reduction, and it’s unfortunate that it hasn’t been studied more for these applications, and pharmaceutical-grade products have not been developed.

References:  Some variety in the way the references are typed out.  Some have periods at the end of abbreviations in the journal title (esp near the beginning) others do not. 

31:  incomplete

32, 42, 43, 45, 48, 51, 53, 58, :  Different capitalization scheme in the article title compared to other references.

37:  Didn’t format correctly (indentation wrong)

Author Response

See attached Word document.

Reviewer 2 Report

The review article is interesting in it's concept and deals with very important issue which is Lead toxicity , but it needs major improvement to be suitable for publication.

1- Introduction part need more reinforcement by more information , authors can use the following article that can help them:

Antioxidant Effect of Carnosine on Aluminum Oxide Nanoparticles (Al2O3-NPs)-induced Hepatotoxicity and Testicular Structure Alterations in Male Rats.

DOI: 10.3923/ijp.2018.740.750

2- Authors must add table with the most relevant risks of lead with articles information

3- It is important to add graphical abstract for mechanisms of lead toxicity to be clear fro readers.

4-Figure 2 need to be professionally plotted.

5-Authors must add conclusion part to draw attention of readers with the most important issues of the review.

Author Response

See attached Word document.

Round 2

Reviewer 1 Report

P4, L 153:  DEA was spelled out on P2, L59, so parenthesis don’t need to be used for the initials here. FDA was first used on P2, L80 of text and should be spelled out there with the initials in parenthesis, from there on the initials can be used without parenthesis.

P4, L165: I found a reference that suggests substitution of other plant material in a kratom sample: https://doi.org/10.1016/j.forsciint.2013.09.016

P4, L191; P6, L227:  Not really obvious, since the FDA limits are supposed to be well below the toxic threshold.  I generally dislike the use of the term “obvious” in the scientific community, because what one person considers obvious, another may not, depending on their background and experience.

P4, L198-201:  I do hope you double-checked the math for these values!

Author Response

See Word document

Reviewer 2 Report

Authors didn't respond to most of comments , some points need to be corrected and improved:

1- The authors  need to reinforce introduction part by paragraph in oxidative stress that resulted from lead contamination and generally mechanism of oxidative stress inducers , so I put some DOI that can help authors about oxidative stress and antioxidant enzymes from 1st time , for example as:

Possible Ameliorative Effects of the Royal Jelly on Hepatotoxicity and Oxidative Stress Induced by Molybdenum Nanoparticles and/or Cadmium Chloride in Male Rats ( https://doi.org/10.3390/biology11030450).

2- I asked authors to add table for summarization of the most important information about lead contamination and benefits of Kratom Products, it is professional step and easily send the message of your article not  redundant and unnecessary.

Author Response

See attached Word document

Round 3

Reviewer 2 Report

OK, changes done

Author Response

See Word document
